# Feeding Ecology of *Odontaster validus* under Different Environmental Conditions in the West Antarctic Peninsula

**DOI:** 10.3390/biology11121723

**Published:** 2022-11-28

**Authors:** Lisette Zenteno-Devaud, Gabriela V. Aguirre-Martinez, Claudia Andrade, Leyla Cárdenas, Luis Miguel Pardo, Humberto E. González, Ignacio Garrido

**Affiliations:** 1Departamento de Ecología, Facultad de Ciencias, Universidad Católica de la Santísima Concepción, Concepción 4081112, Chile; 2Química y Farmacia, Facultad de Ciencias de la Salud, Universidad Arturo Prat, Iquique 1110939, Chile; 3Laboratorio de Ecología Funcional, Instituto de la Patagonia, Universidad de Magallanes, Punta Arenas 6210738, Chile; 4Facultad de Ciencias, Universidad Austral de Chile, Valdivia 5110566, Chile; 5Centro FONDAP de Investigación de Dinámica de Ecosistemas Marinos de Altas Latitudes (IDEAL), Universidad Austral de Chile, Valdivia 5110566, Chile; 6Laboratorio Costero de Recursos Acuáticos de Calfuco (LCRAC), Facultad de Ciencias, Universidad Austral de Chile, Valdivia 5110566, Chile; 7Québec-Océan, Département de Biologie, Université Laval, Québec City, QC G1V 0A6, Canada

**Keywords:** trophic ecology, stable isotope analysis, Antarctic Peninsula, sea star, global warming, Antarctic benthic communities

## Abstract

**Simple Summary:**

Even though researchers have previously examined the diet of *O. validus*, the feeding ecology of this species of sea star is still poorly understood. To address this issue, we used a multifaceted approach to investigate the diet of *O. validus* from three systems with marked environmental differences of the northern Antarctic Peninsula. The results showed a significant divergence of *O. validus* δ^15^N–δ^13^C values among regions, suggesting a habitat-specific foraging behavior and confirming the ability of this species to switch resource utilization across differing habitat compositions, which may be a key survival response in the face of environmental change.

**Abstract:**

To study how *Odontaster validus* can influence the spatial structure of Antarctic benthic communities and how they respond to disturbance, it is necessary to assess potential dietary shifts in different habitats. We investigated the diets of *O. validus* from Maxwell Bay and South Bay in the West Antarctic Peninsula. A multifaceted approach was applied including in situ observations of cardiac stomach everted contents, isotopic niche, and trophic diversity metrics. Results confirm the flexible foraging strategy of this species under markedly different environmental conditions, suggesting plasticity in resource use. The data also showed evidence of isotopic niche expansion, high δ^15^N values, and *Nacella concinna* as a common food item for individuals inhabiting a site with low seasonal sea ice (Ardley Cove), which could have significant ecological implications such as new trophic linkages within the Antarctic benthic community. These results highlight the importance of considering trophic changes of key species to their environment as multiple ecological factors can vary as a function of climatic conditions.

## 1. Introduction

One of the most abundant and conspicuous consumers of Antarctic ecosystems is the red sea star *Odontaster validus* [1,2]. Several aspects of *O. validus* ecology and biology have received attention, including its reproductive cycle [2,3,4,5], diet composition [6,7,8,9,10], population structure [2], and threats, such as disease [11] and increasing temperature [12,13,14,15]. Nonetheless, detailed data on the effects of food availability and trophic interactions are still scarce [5,16].

*O. validus* is a key species in Antarctic benthic communities [6,10], where it can exert different trophic roles [1,2,9] with different foraging tactics (i.e., omnivore, filter feeder, scavenger, herbivore, active predator, necrophagous, spongivore), allowing individuals to forage over a wide range of prey, including diatoms, amphipods, isopods, sponges, gastropods, bivalves, hydroid, sea star, sea urchin, detritus, nauplii, and algae [1,3,5,6,7,8,9]. Therefore, disentangling feeding behavior and associated trophic relationships is crucial to investigate the nature and magnitude of interactions in top-down and bottom-up regulation and to infer specific responses to habitat modification [17,18,19].

Stomach contents are often used to determine relative composition of prey species, but this method can be misleading because it does not necessarily describe what organisms can assimilate [20]. Analyses of stable isotopes of carbon and nitrogen is a powerful tool to provide time-integrated information on diet and habitat use. Stable isotopes of carbon and nitrogen (^13^C/^12^C, denoted as δ^13^C) vary across ecosystems and have the potential to trace biogeochemical processes and trophic interactions [21,22,23]. δ^13^C values distinguish the origins of organic matter [24], whereas the δ^15^N values are useful as a tracer for trophic levels [25]. It has been proposed that the variability of isotopic composition of a population or a species (i.e., its isotopic niche) can be used as a proxy to assess the trophic niche and, thus, the degree of individual specialization in consumers [26,27]. Advances in statistical analyses have made it possible to obtain a quantitative analysis of trophic ecology by metrics. More precisely, these metrics provide empirical and conceptual examples of trophic diversity, redundancy, and vertical and basal trophic structure [28,29].

Habitat modification of coastal Antarctic benthic communities is becoming increasingly common due to global warming [30,31,32]. *Odontaster validus* is highly dependent on benthic habitats as it responds rapidly to any kind of food falling to the seafloor, usually ingesting the entire food source within hours [19]. In fact, recent records show isotopic niche constriction in *O. validus* populations from sites with high activities of glacier meltwater and ice disturbance [16], as well as changes in size, coloration, nutrition, and reproductive effort as a result of primary production variability [2]. Accordingly, the foraging ecology of *O. validus* is closely linked to environmental conditions [2,15,16]; therefore, one might expect markedly differences in feeding ecology associated with context-specific habitat, especially around the northern Antarctic Peninsula where seasonal sea ice is dynamic and the distribution of resources is uneven [33,34]. 

Here, we quantitatively examined isotopic niches, trophic structure variation and the relative contribution of potential prey items in specimens collected from three systems with marked environmental differences of northern Antarctic Peninsula (low seasonal sea ice, high sea ice cover, and high activities of glacier meltwater and ice disturbance) to test hypotheses about *O. validus* dietary variation among regions. Additional analyses such as the relationships between disk radius and δ^15^N–δ^13^C values and composition of stomach contents were done to complement this study.

## 2. Materials and Methods

### 2.1. Study Sites

This study was conducted at two sites in Maxwell Bay (Marian Cove and Ardley Cove) on King George Island (South Shetland Archipelago), and one site on South Bay in Doumer Island (Palmer Archipelago) (Figure 1). Details of study sites are shown in Table 1.

#### 2.1.1. Marian Cove—High Activity of Glacier Meltwater and Ice Disturbance

Marian Cove is a rapidly warming and deglaciating small fjord mall (4.5 km in length, 1.5 km in width, and up to 120 m deep) where tidewater glaciers have retreated approximately 1.9 km over the last six decades [35]. Two zones can be distinguished in there including an inner cove, where maximal depths can reach 50 m (average = 30 m), and an outer deeper cove (>100 m) [35,40]. The present study focused on inner cove (0.2 km to glacier), where glacier break-up occurs throughout the summer months, introducing large volumes of turbid meltwater and icebergs into the cove [35,41,42,43]. Previous studies on the benthic faunal assemblages revealed a variety of filter feeders, including ascidians, sponges, and polychaetes associated to benthic diatom blooms [44,45].

#### 2.1.2. Ardley Cove—Low Seasonal Sea Ice

Ardley Cove is a small area (2.5 km in length, 1.5 km in width, and up to 50 m deep) located in the innermost part of the Maxwell Bay (Figure 1A). Local hydrography is influenced mainly by marine water [36]. The benthic community of Ardley Cove is characterized by highly abundant and diverse macroinvertebrates and macroalgae, with a subtidal zone dominated by brown and red microalgae, several species of fishes, scavengers, grazers, sponges, bryozoans, and ascidians [46,47].

Another important characteristic in the study area is a physical and biogeochemical gradient between Ardley Cove and Marian Cove because of decreasing concentrations of suspended particulate matter and increasing temperature and salinity values with increasing distance from the meltwater sources [48,49].

#### 2.1.3. South Bay—High Sea Ice Cover

South Bay is a small bay (2.6 km in length, 1 km width, and up to 222 m deep), largely covered by an ice cap with small subglacial discharge outflows and some ice calving around the fjord periphery [37]. Benthic habitats are characterized by rocky steep slopes and a marked shift from motile fauna at shallower areas (<10 m), to large sessile benthos at depths down to 30 m. Sites down to about 40 m depth appeared to be strongly affected by ice scouring, resulting in the occurrence of a marked patchiness in the distribution of communities [50]. 

### 2.2. Field Sampling

#### 2.2.1. *O. validus*

Seventeen adults (arm length > 22 mm [1]) of *O. validus*, were collected per site during the austral summer 2017/2018 from subtidal sites (5–20 m depth). In Marian Cove, individuals were collected by hand whereas in Maxwell Bay, suction dredge sampling was the method of choice using a portable underwater venture suction dredge device equipped with a 20-µm mesh size. All organisms were measured to record arm length (distance from the mouth to the tip of the longest arm) and then sorted and frozen (−20 °C) until it was time to analyze them. 

Four transects running perpendicular from the shore to a depth of 15 m were also surveyed to observe the in situ stomach contents of 15 adult *O. validus*. These observations were performed during 30-min surveys by SCUBA divers. Within each time interval, a total of 15 specimens were measurement and we flipped over every one to record if it was feeding (i.e., with stomach everted and a prey trapped within) and, if so, we wrote down which prey species were being consumed.

#### 2.2.2. Potential Prey

In subtidal sites (depth < 30 m) at Ardley Cove and South Bay specimens of *Gondogeneia antarctica*, *Nacella concinna,* and *Palmaria decipiens* were collected by hand by SCUBA divers during the austral summer of 2018. The specimens were washed repeatedly with milliQ water and dried at 60 °C for 48 h. 

#### 2.2.3. Suspended Particulate Organic Matter (POM)

Water column POM data were obtained from water samples taken with 10-L go-flo bottle at 5-m during the austral summers of 2017 at Ardley Cove and South Bay. Water samples were filtered with GF-F glass fiber filters (47 mm diameter, 0.7 μm pore size), treated with 1M HCl, and rinsed with distilled water to eliminate carbonates. 

### 2.3. Stable Isotope Analysis

In the laboratory, mollusk mantle, crustaceans (entire), and the tegument of each arm of *O. validus* were thawed with distilled water and oven-dried at 60 °C for 36–48 h, and then ground to a fine powder with a mortar and pestle. As sea stars’ and crustaceans’ endoskeletons are made of carbonates which, in turn, are more enriched with ^13^C than other tissue components, these tissues were acidified for the removal of carbonates by wetting the filters with 1 mL 0.5M hydrochloric acid (HCl). Lipids were extracted from all tissues with a chloroform/methanol (2:1) solution [51]. This was done because, compared to other molecules, lipids are depleted of ^13^C [52] and their concentration in tissues may vary between and within species.

Approximately 0.3–0.5 mg of tegument, mollusk mantle, and crustaceans (entire) were weighed in sterilized tin capsules and analyzed for dual stable carbon and nitrogen isotopes. All the consumers and benthic primary food sources were analyzed at the Pontificia Universidad Católica de Chile (Flash EA200 IRMS Delta Series, Thermo Scientific, Bremen, Germany), and the suspended POM was analyzed at the University of California, Davis, using a PDZ Europa ANCA-GSL elemental analyzer interfaced to a PDZ Europa 20–20 isotope ratio mass spectrometer (Sercon Ltd., Cheshire, UK). 

Stable isotope ratios were reported in standard (δ) notation using the following equation:
δX = [(R_sample_/R_standard_) − 1] × 1000(1)
where X is either ^13^C or ^15^N, R is the ratio of heavy to light isotopes, and the standard is either PDB for ^13^C or atmospheric N^2^ for ^15^N. The reproducibility, as determined by the standard deviation of the internal standards, was ±0.18‰ and 0.30‰ for δ^15^N and δ^13^C, respectively.

### 2.4. Stomach Contents Analysis

The importance of the different prey items was evaluated calculating the frequency of occurrence (%F = number of stomachs containing prey i/total number of stomachs containing prey × 100) [53]. Prey were identified to the lowest possible taxonomic level on the basis of their digestion state.

### 2.5. Statistical Analysis

Permutational multivariate analysis of variance (PERMANOVA) [54,55], based on Bray–Curtis dissimilarity matrix (estimated probabilities were based on 999 permutations), was used to examine the divergence of *O. validus* δ^15^N–δ^13^C values among regions. A posteriori pairwise comparisons were used to determine significant differences among sites.

Absolute values of δ^13^C were used and all data were square root transformed prior to analyses to reduce the effects of outliers. We performed these analyses using the software Primer-7 [56]. An alpha level of 0.05 was used throughout to indicate statistical significance.

Trophic structure, in terms of trophic diversity and trophic redundancy of *O. validus* populations from the different regions, was investigated by calculating six quantitative community-wide Layman metrics using mean carbon and nitrogen isotope ratios of the consumers [29]. Trophic diversity is reflected by (1) δ^15^N range (NR) which represents the distance from the most depleted to the most enriched δ^15^N values between species and provides information regarding vertical trophic structure, (2) δ^13^C range (CR) is the distance from the most depleted to the most enriched δ^13^C values between species and represents the niche diversification with respect to the food sources; higher CR reflects the utilization of a broader spectrum of food sources; and (3) mean distance to centroid (CD), is the average Euclidean distance of each species to the centroid and provides information about the average degree of trophic diversity. Trophic redundancy, then, is characterized by (4) the mean nearest neighbor distance (MNND), which is the mean Euclidean distance in the bivariate isotopic space of each specimen to its nearest neighbor, and as such reflects a measure of overall density of species packing; (5) the standard deviation of nearest neighbor distance (SDNND), which is calculated as the standard deviation of these Euclidean distances and is a measure of the evenness of the spatial density and packing of the specimens in the isotopic space. Low MNND values (increased trophic redundancy) are an indication of populations with a large proportion of specimens with similar trophic ecologies, while low SDNND values suggest more even distribution of the isotopic niches [29]. In addition, both SEAC (the corresponding metric corrected for small sample size) and the Bayesian estimation of ellipse size SEAB were calculated to, respectively, represent the total niche space and to probabilistically compare the isotopic niches between regions [57,58]. Bayesian estimations of the isotopic niche and Layman metrics (based on 10^5^ successive iterations) were computed for each site and subsequent pairwise comparisons were computed with the SIBER package [27,57]. Pairwise comparisons of metrics between sites were calculated by comparing pairs of draws from the posterior distributions. If this percentage exceeded 95% or it was lower than 5%, the differences between metrics were considered as meaningful. All computations of metrics were done in R version 3.2.1. Finally, to estimate the relative contribution of prey items to *O. validus,* in each study site, we used the upgrade function simmr from the package Stable Isotope Mixing Models within R. This upgraded version contains a sophisticated mixing model that allows realistic assessments and discerning source contributions to a mixture [58]. Models were built over four Markov chains with 10,000 steps per chain and a burn-in of 1000 iterations. Data within each group fitted a normal distribution [59]. The model included the fractionation factor (Δ) suggested by McCutchan et al. [60] for different species (Δ^13^C, mean ± SD = 0.4 ± 1.2‰; Δ^15^N = 2.3 ± 1.6‰).

Potential prey items were selected according to studies analyzing stomach contents [1,3,5,6,7,8,9] and this study], although they may not give full coverage of the diet due to biases in sampling. δ^15^N–δ^13^C values of prey items from Marian Cove were obtained from Ahn et al. [61] (*G. antarctica*, *P. decipiens,* and suspended POM) and Choy et al. [62] (*N. concinna*).

Statistical analyses included a Kolmogorov–Smirnov test with the Lilliefors correction to explore whether *O. validus* δ^15^N–δ^13^C values followed normal distributions. Additionally, lineal regressions were used to examine how δ^15^N–δ^13^C values varied with the disk radius. These analyses were carried out with PASW Statistics (version 17.0 for Windows, SPSS, Chicago, IL, USA). Significance was assumed at 0.05.

## 3. Results

### 3.1. Divergence of O. validus δ^15^N–δ^13^C Values among Regions

A summary of PERMANOVA test and post hoc pairwise comparisons are shown in Table 2. We found significant differences in *O. validus* δ^15^N–δ^13^C among sites (Table 2; Figure 2). *O. validus* ^13^C values from South Bay was usually enriched in ^13^C compared with the individuals from Marian Cove and Ardley Cove (Figure 2A). Furthermore, in Ardley Cove *O. validus* were characterised by isotopic signatures enriched ^15^N (Figure 2B). Pairwise tests showed that *O. validus* δ^13^C values varied significantly between Marian Cove and Ardley Cove and South Bay and Marian Cove (Table 2). 

### 3.2. Isotopic Niche and Trophic Structure of O. validus Populations among Regions

The generation of standard ellipses with the mean station-corrected δ^13^C and δ^15^N values of *O. validus* (Figure 3A) and the subsequent computation of SEAB (Figure 3B) showed that Ardley Cove *O. validus* population had the largest SEAB (mode = 0.6‰2, 95% credibility interval CI95 = 4.4–12.0‰2) followed by Marian Cove *O. validus* population (mode = 2.8‰2, CI95 = 1.6–4.6‰2). South Bay *O. validus* population had the lower SEAB (mode = 2.3‰2, CI95 = 1.3–3.9‰2). Overall, the trophic diversity (CR, NR, CD) and the trophic redundancy (MNND and SDNND values) differed among regions (Figure 4, Table 3). The largest trophic diversity (CR~8.07, NR~3.93 CD~2.55) and absolute trophic redundancy values (MNND~0.66 and SDNND~0.42) were found in *O. validus* population from Ardley Cove, showing that this zone is characterized by a high trophic diversity and low trophic redundancy.

In contrast, the population from South Bay had the lowest absolute trophic diversity (CR~6.90, NR~1.3, CD~1.52) (Figure 4A–C) and redundancy values (MNND~0.52 and SDNND~0.29) (Figure 4D,E). The probabilities that the metrics of Ardley Cove were larger than those from other zones generally exceeded 0.7 (Table 3). 

### 3.3. Exploring O. validus δ^15^N–δ^13^C Values with Disk Radius 

Individuals from both Marian Cove and Ardley Cove regions showed positive linear relationships between disk radius and δ^15^N–δ^13^C values (Figure 5) indicating a shift in dietary source and trophic level with size, but no relation was found for *O. validus* from South Bay (Figure 5C,D).

### 3.4. Observing In Situ Cardiac Stomach Everted Contents of O. validus from Ardley Bay

All cardiac stomachs examined (*n* = 12) contained food items. A total of 9 different taxa and 6 species could be positively identified (Figure 6). The gastropod *N. concinna* was the main prey observed in stomachs (Fo = 34%), followed by amphipods (Fo = 17%) and *Limatula hodgosoni* (Fo = 13%). Polychaetes, *Sterechinus neumayeri*, *Margarella antarctica*, Littorinids, and the seastar *Neosmilaster georgianus* also constituted the *O. validus* diet.

### 3.5. Relative Contribution of Four Potential Prey to O. validus Diet

Overall, the simmr model output showed that different contributions of *N. concinna*, suspended POM, *G. antarctica,* and *P. decipiens* to *O. validus* diet across regions. *O. validus* at Marian Cove derived a high proportion of their diets (>60%) from the amphipod *G. antarctica* (Figure 7A). Individuals from South Bay showed a similar proportion of food sources, although *N. concinna* was slightly larger (Figure 7B). Conversely, *O. validus* at Ardley Cove derived over 40% of its isotopic signature from *N. concinna* and *G. antarctica* (Figure 7C). Details of *O. validus* stable isotope ratios and their potential prey are shown in Table 4 and Figure 7D–F. 

## 4. Discussion

Our work shows a significant divergence of *O. validus* δ^15^N–δ^13^C values among regions, which confirms the flexible foraging strategy of this species under markedly different environmental conditions. These findings may have implications for food web models, as patterns of resource use documented among specimens from one site may not be applicable to a neighboring location. 

Previous studies suggested important spatio-temporal dynamics of sea-ice cover related variations of basal resource inputs [15,63,64,65,66]. For instance, in regions where the persistence of sea ice is high, sympagic pelagic producers may support benthic organisms [15], whereas in regions free of sea ice most of the year, a mix of ice-derived materials, phytoplankton blooms and benthic sources occurs and sustains nearshore benthic communities with variable degrees of benthopelagic coupling [63,67]. The divergence of *O. validus* δ^15^N–δ^13^C values among regions reported here seems consistent with these findings, as high δ^13^C values and low δ^15^N values of *O. validus* from South Bay suggest that sympagic algae may be an important food source in an area where high sea ice concentration is present and where there is a long sea ice season. On the other hand, lower δ^13^C and higher δ^15^N values of *O. validus* from Maxwell Bay may be related to massive diatom blooms and high primary production rates recorded in the region [39] as phytoplankton blooms induce decreasing δ^13^C values in POM [68,69] and increase ^15^N values of surface waters as a result of the preferential consumption of ^14^N by phytoplankton. However, it is important to note that the high variability in these values, mainly in Ardley Cove, also suggest that *O. validus* assimilate a mixture of detrital and fresh organic matter derived from phytoplankton and macroalgae material, which is corroborated by Zenteno et al. [67].

The isotopic niche of *O. validus* from Ardley Cove was widest (also reflected by large NR and CR), likely because of high diversity of prey species in a region known for its complex trophodynamics and different primary food sources [36,67]. Similarly, detailed investigations in the outer section of Ezcurra Inlet (King George Island) show the widest isotopic niche of *O. validus* on a site with high benthos diversity. This corroborates current knowledge that the *O. validus* is a food generalist [1,2,9]. Furthermore, *O. validus* from Ardley Cove had the highest mean δ^15^N values and high reliance on sessile and mobile prey, indicating a prevalence of omnivory. Omnivory is a feeding strategy in which a consumer feeds at multiple trophic levels [70], which leads to an increase in connectance by adding weak links [71]. Theoretical and empirical evidence suggests that omnivory can stabilize food webs when trophic interactions are weak [71,72,73]. Consequently, the prevalence of omnivory of *O. validus* could have significant ecological implications within the Antarctic benthic community. 

While the dietary niches of *O. validus* narrowed in Marian Cove and South Bay when compared to those of individuals in Ardley Cove, the degree of narrowing varied between both sites. In that respect, the isotopic niche width at South Bay was smaller than the isotopic niche width at Ardley Cove and Marian Cove, indicating a more restricted use of resources. This pattern is corroborated by the low CR values and a similar contribution of potential prey in individuals of South Bay. A more restricted use of resources may be context specific. For instance, in some regions, heterotrophic resources may be homogeneously distributed through the water column due to unique oceanographic regimes [74], which could be adjusting the proportion of each prey in a mixed diet with similar isotopic compositions and, thus, go undetected on bulk stable isotope analyses on a site where prey are predictable both temporally and spatially. 

Isotopic niche width at Marian Cove was smaller than isotopic niche width at Ardley Cove, reflecting a more restricted use of resources and limited inter-individual differences, which is further confirmed by the low Layman metrics values (CR, CD and MNND). A similar pattern was observed in the inner Ezcurra Inlet Admiralty Bay (King George Island), where low trophic niche constriction occurs in a site with high turbidity, which could be associated with competition avoidance in a resource-limited system and intraspecific specialization of this species [16]. Interestingly, *O. validus* from Marian Cove feed opportunistically on mobile prey (*G. antarctica*), which could be linked to lower sessile prey availability due to the activities of glacier meltwater and ice disturbance in the study area [35,41,75,76]. These observations are consistent with what echinoderms do with ice disturbances, which has been recorded through monitoring carried out in other areas such as McMurdo Station [77]. 

Bayesian modeling offers a novel framework for quantifying the trophic plasticity and identifying the trophic strategies of *O. validus*. Specifically, our assessment of the contribution of prey is based on the assumption that the analysis of a relatively small number of items can be extrapolated to build generalizations of trophic ecology [67]. This approach can lead to biased conclusions because of a low sampling effort of other potential prey items (e.g., isopods, sponges, hydroid, sea star, sea urchin, detritus). Consequently, further work should be conducted to better understand the percentage of prey assimilation using multiple potential preys across regions, as well as the sources of variability to successfully apply this method within highly diverse systems.

Finally, size-based shifts in δ^13^C and δ^15^N values were evident in Marian Cove and Ardley Cove, with the increasing δ^13^C and δ^15^N values across disc radius. This shift can be explained by the replacement of smaller prey occupying lower trophic levels by larger and/or more carnivorous benthic prey in the diet during growth [16] and the ability to feed through different pathways in sites with different subsidies of basal resources. Unexpectedly, this shift was not observed in individuals from South Bay, possibly reflecting uniform foraging strategy regardless of age or a homogeneous supply of basal resources. These results show that *O. validus* is an omnivore that may modify its diet based on the profitability and abundance of its resources, either by switching prey or by adjusting the proportion of each in a mixed diet.

## 5. Conclusions

In summary, our work shows *O. validus* dietary variation among regions, suggesting that habitats shape potential dietary preferences in this species. Additionally, the high δ^15^N values and the prevalence of *N. concinna* as common food items for individuals inhabiting Ardley Cove may indicate that low seasonal sea ice durations directly influence *O. validus* trophic ecology, inducing a carnivore behavior, which in turn, may force the community to new trophic linkages. The ability of *O. validus* to vary resource utilization across differing habitat compositions may be a key response allowing their persistence and survival in the face of environmental change. However, fluctuations in resource availability and environmental changes could increase *O. validus’* dietary complexity with implications for associated species. 

## Figures and Tables

**Figure 1 biology-11-01723-f001:**
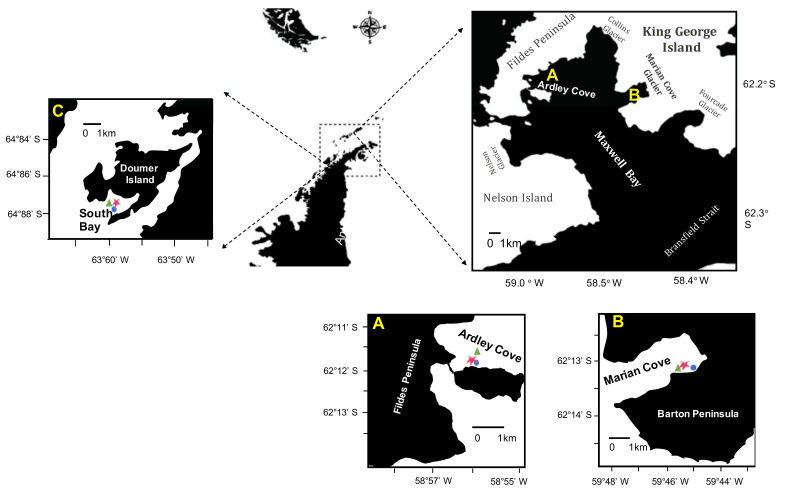
Site locations where *O. validus*, prey, and the suspended particulate organic matter were sampled for stable isotopes analyses. Red circles indicate the three sites of study in the Western Antarctic Peninsula. Sea star denote sample sites of *O. validus*, triangles denote prey sites, and blue circles shows the suspended POM sites. Upper panels display sampling stations in Maxwell Bay (**A**,**B**) and South Bay (**C**). Lower panel display sampling station in Ardley Cove (**A**) and Marian Cove (**B**).

**Figure 2 biology-11-01723-f002:**
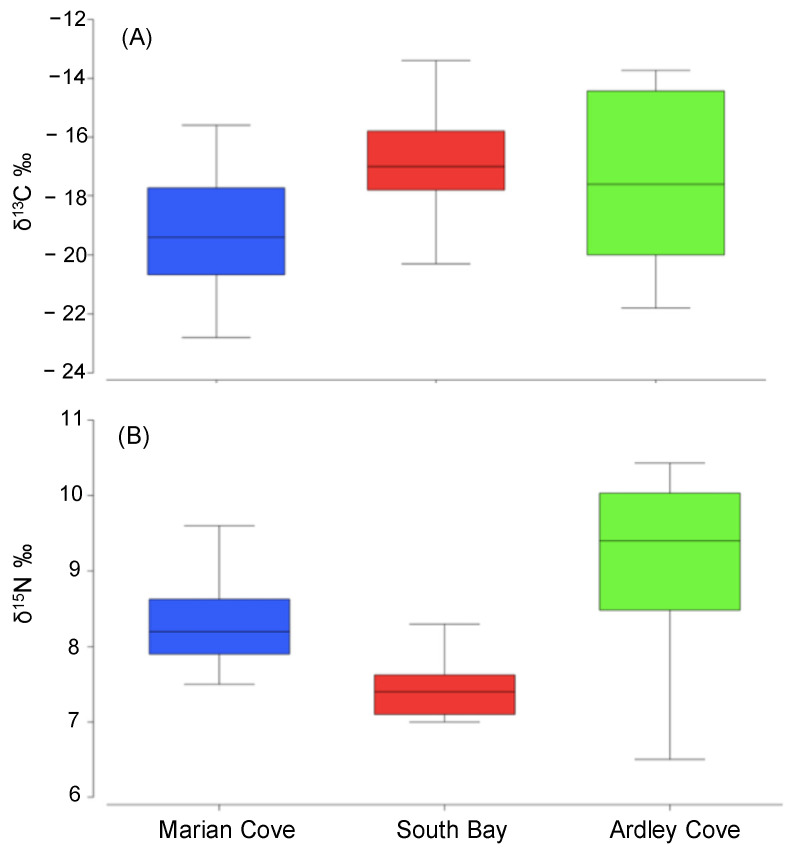
Box-whisker plots showing variation in *O. validus*–δ^13^C (**A**) and δ^15^N (**B**) values among regions.

**Figure 3 biology-11-01723-f003:**
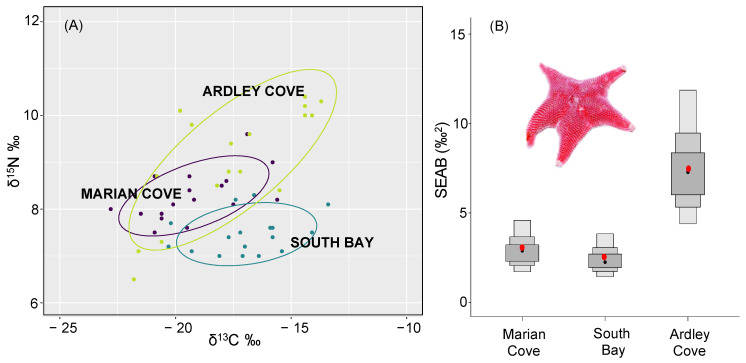
In the left panel, (**A**) solid lines enclose the standard ellipse areas corrected for the sample size (SEAc) of *O. validus* populations from Ardley Cove, Marian Cove and South Bay. The right panel (**B**) shows the standard ellipse area Bayesian estimations (SEAB) for each population. Black dots show the SEAB mode, red dots represent the SEAC, and gray boxes the probability of data distribution (50% dark grey boxes, 75% light grey boxes and 95% lighter grey boxes).

**Figure 4 biology-11-01723-f004:**
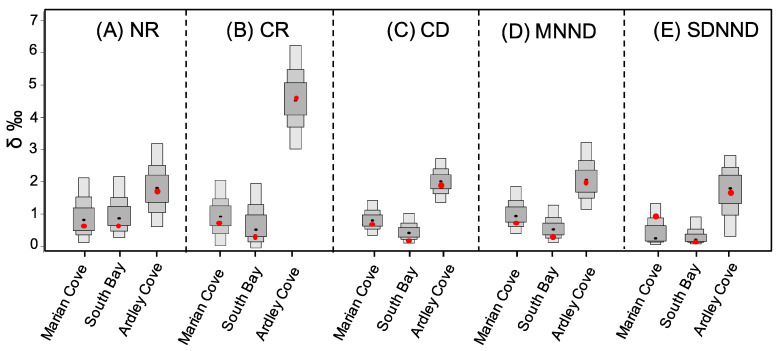
Density plot showing the 95, 75, and 50% credible intervals of Layman’s metrics in Marian Cove, South Bay and Ardley Cove using Bayesian techniques. Black dots represent the mean standard for each metrics; red dots indicate the mean of corrected metrics.

**Figure 5 biology-11-01723-f005:**
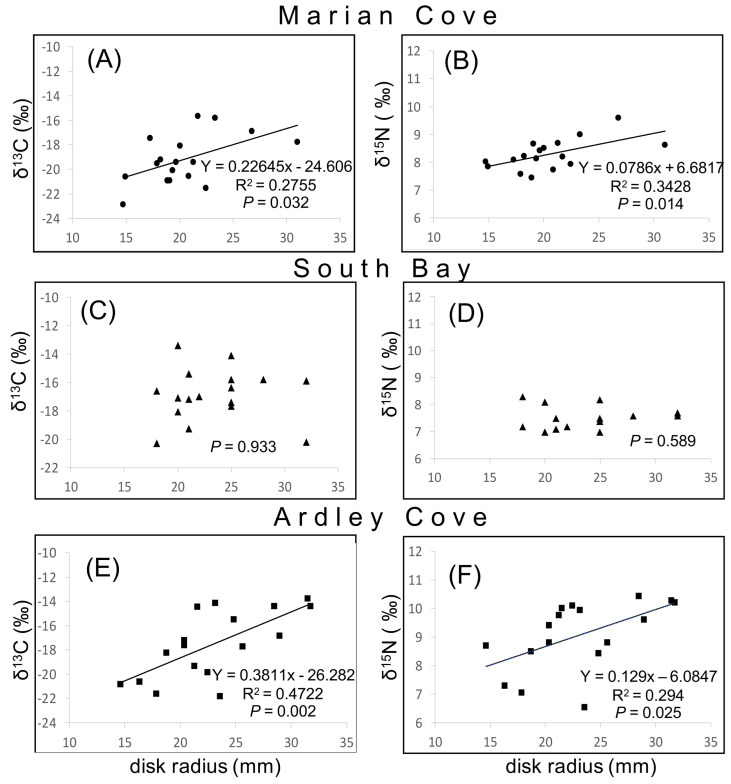
Correlation coefficients between disk radius and δ^13^C and δ^15^N of *O. validus* from Marian Cove (**A**,**B**), South Bay (**C**,**D**), and Ardley Cove (**E**,**F**).

**Figure 6 biology-11-01723-f006:**
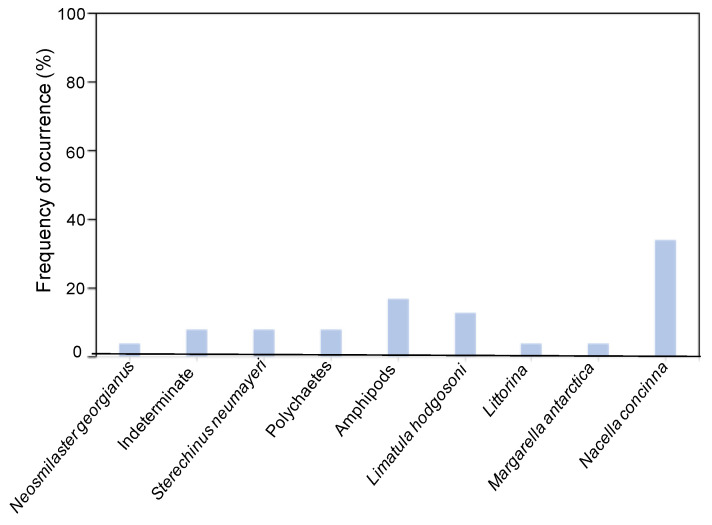
Frequency of occurrence of *O. validus* cardiac stomach everted contents from Ardley Cove.

**Figure 7 biology-11-01723-f007:**
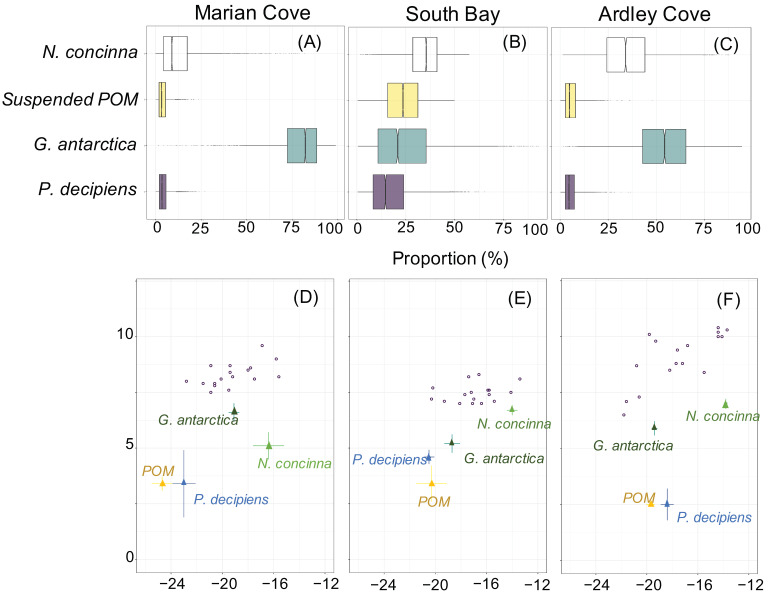
The top panel (**A**–**C**) presents the dietary composition of *O. validus* across regions according to simmr model. Boxplots represent 95% credible interval of primary food sources assimilation for each consumer. The centerline in the box is the median of all solutions, and the box is drawn around the 25% and 75% quartiles, thereby representing 50% of the solutions. The δ^13^C and δ^15^N values of consumers were corrected for the discrimination factor so that they could be compared to the prey values. In the bottom panel, bivariate stable isotope ratios of *O. validus* and their potential prey items from Marian Cove (**D**), South Bay (**E**), and Ardley Cove (**F**) are shown without the correction of the discrimination factor.

**Table 1 biology-11-01723-t001:** Details of study sites. Salinity and seawater temperature gradient refer to surface waters (<10 m depth) throughout the year.

	Maxwell Bay		South Bay
	Marian Cove	Ardley Cove	
Salinity gradient (psu)	33.8–34.1 [35]	33.97–34.14 [36]	34.2–34.7 [37]
Seawater temperature (°C)	−1.8–1.5 [35]	−1.4–1.8 [36]	−1.7–3.0 [38]
Physical disturbances (runoff)	High [35]	Low [36]	Low [37]
Total chlorophyll maxima (mg Chl-*a* m^−3^)	29.2 [39]		19.7 [39]
Annual sea ice extent	June–July [37]		June–November [37]

**Table 2 biology-11-01723-t002:** Summary of PERMANOVA test and post hoc pairwise comparisons.

**PERMANOVA Test**			
	*F*	*p*	
δ^15^N	16.695	0.001	
δ^13^C	4.5288	0.016	
**Post Hoc Pairwise Comparisons**			
	Group	*t*	*p*
δ^15^N	Marian Cove, South Bay	5.0445	0.001
	Marian Cove, Ardley Cove	2.3037	0.034
	South Bay, Ardley Cove	5.1102	0.001
δ^13^C	Marian Cove, South Bay	3.3621	0.002
	Marian Cove, Ardley Cove	2.0963	0.037
	South Bay, Ardley Cove	0.6094	0.560

**Table 3 biology-11-01723-t003:** Probability that the Layman metric and isotopic niche width (SEAB) of the row is smaller than that of the column.

**δ^13^C Range (CR)**			
	Marian Cove	South Bay	Ardley Cove
Marian Cove		0.8870	0.0088
South Bay	0.1130		0.0018
Ardley Cove	0.9913	0.9983	
**δ^15^N Range (NR)**			
	Marian Cove	South Bay	Ardley Cove
Marian Cove		0.5418	0.2105
South Bay	0.45825		0.1853
Ardley Cove	0.7895	0.81475	
**Mean Distance to Centroid (CD)**			
	Marian Cove	South Bay	Ardley Cove
Marian Cove		0.8373	0.0009
South Bay	0.1628		
Ardley Cove	0.9913	0.9968	0.0033
**Mean Nearest Neighbor Distance (MNND)**			
	Marian Cove	South Bay	Ardley Cove
Marian Cove		0.7988	0.0365
South Bay	0.2013		0.0085
Ardley Cove	0.9635	0.9915	
**Standard Deviation of the Nearest Neighbor Distance (SDNND)**			
	Marian Cove	South Bay	Ardley Cove
Marian Cove		0.6910	0.0943
South Bay	0.3090		0.0470
Ardley Cove	0.9058	0.9530	
**Standard Ellipse Area Bayesian Estimations (SEAB)**			
	Marian Cove	South Bay	Ardley Cove
Marian Cove		0.6935	0.0028
South Bay	0.3065		0.0013
Ardley Cove	0.9973	0.9988	

**Table 4 biology-11-01723-t004:** Summary of the mean and standard deviation (mean ± SD) values of potential prey items and *O. validus* from each study site.

Study Site	Species/Source	δ^13^C (‰)	δ^15^N (‰)	*N*
Marian Cove	*P. decipiens* [61]	−23.1 ± 0.9	3.3 ± 0.8	2
	*G. antarctica* [61]	−19.2 ± 0.4	6.5 ± 0.4	6
	*N. concinna* [62]	−16.5 ± 1.5	5.0 ± 0.6	27
	*Suspended POM* [62]	−24.8 ± 1.2	3.3 ± 0.3	2
	*O. validus*	−19.2 ± 2.0	8.3 ± 0.5	17
South Bay	*P. decipiens*	−20.6 ± 0.4	4.5 ± 0.3	5
	*G. antarctica*	−18.8 ± 0.6	5.1 ± 0.4	5
	*N. concinna*	−14.1 ± 0.4	6.6 ± 0.2	5
	*Suspended POM*	−20.4 ± 1.2	3.3 ± 0.8	4
	*O. validus*	−16.9 ± 1.9	7.5 ± 0.4	17
Ardley Cove	*P. decipiens*	−18.5 ± 0.5	2.4 ± 0.0	5
	*G. antarctica*	−19.5 ± 0.2	5.8 ± 0.3	5
	*N. concinna*	−13.9 ± 0.7	6.9 ± 0.2	5
	*Suspended POM*	−19.8 ± 0.1	2.4 ± 0.0	2
	*O. validus*	−17.5 ± 2.8	9.1 ± 1.2	17

## Data Availability

The data presented in this study are available upon request from the corresponding author. The data are not publicly available because of restrictions related to privacy rights.

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
