# Peer review of "Feeding Ecology of Odontaster validus under Different Environmental Conditions in the West Antarctic Peninsula"

_biology, 2022, doi:10.3390/biology11121723_

Round 1

Reviewer 1 Report (New Reviewer)

 Review of ‘Feeding ecology of Odontaster validus under different environmental conditions in the West Antarctic Peninsula’ (Zenteno-Devaud et al)

In this manuscript, the authors report on research on the trophic ecology of Odontaster validus, an important species in the Antarctic Peninsula. The design is very elegant: different areas, subjected to different environmental conditions were visited, and the species of interest was investigate using stable isotope techniques and observation of feeding animals. They reveal that O. validus is indeed capable of adapting its diet, depending on the environmental condition. One the hand, this is important information to be taken into account when generalizing findings, one the other hand, the information is important in the light of climate change. The text is generally well-structured and well-written, but at some place a more in-depth discussion is required. I recommend to accept this manuscript, after some detailed comments (below) are taken into account:

 Line 27: species name should be in italic

Line 120: Polychaetes can be written with small 'p': polychaetes.

Line 166: it is not clear whether 15 individuals were flipped over, or whether enough individuals were flipped over until 15 observations on stomachs were recorded.

Line 209-210; it is unclear what is meant with 'the level of spread'? This needs to be clarified.

In addition, I am surprised to see a combination of MDS (wich is generally a NON-metric multidimensional scaling in PRIMER). The MDS algorithm is different from the PERMANOVA algorithm. I suggest to use PCO in PRIMER, as this visualisation technique uses an algorithm that is more similar to PERMANOVA.

 Line 214: I find it strange to do spend a lot of energy (and probably money) to collect stable isotope values of organisms from 3 different regions, and then transform them. The authors could go for an alternative approach and work with the real data, and provide (or speculate) on the reasons for the outliers. Often, there is valuable ecological information in such values.

 Line 235: I believe that the correct notation for SEAC is with the "C" in subscript. Please check notation for SEAB as well.

 3.1. This section is confusing, and I have the feeling that is over-analysed. I agree that it makes sense to check if the 13 C values, and the 15 N values are different between regions. Hence, a PERMANOVA for 13C, and a PERMANOVA for 15N with a visualisation by box-whisker plots would be sufficient. The MDS, based on 2 variables is not needed (and doesn't make sens, as 2 variables can be visualised using a x-y plot?

 Line 263. I don't understand why there would be 'more spatial structure' in the values from South Bay. The graph suggests that the values from South Bay are different from the values recorded in the other 2 locations.

 Line 264-266: this can be written easier.? I guess the bottom line is that the pairwise tests indicated that the values in Marian Cove are significantly lower than in the other 2 areas?

 Line 268-271: A rather lenthy description to mention that the 15N values are significantly different between regions. I actually suggest to bring all the results of the PERMANOVA (general and pairwise test) in a table, and describe the graphs. This will shorten the text, and bring clarity for the reader.

 Line 291 - 293: this is difficult to see in the figure. It would make more sense to have panels per 'index', and three values (for each region) per panel. Then, the reader doesn't need to check different panels for each 'index'.

 Line 300: Table 2?

 Line 332-333: this approach is not described in the Material and Methods? It is a linear regression, hence should be mentioned there.

 Line 336: this is actually discussion (and the authors overlooked the possibility that different prey items can have similar 13 C values)

 Line 426: Another explanation can be 'no specialisation at all', and hence the capacity to feed on very different food items?

 Line 429-430: this needs some more explanation. Why are these new trophic linkages, and why is this very important?

 Line 445-447: it is nice to see that there are other areas with lowered isotopic niche width, and an associated explanation. However, there is no explanation provided for the lower isotopic niche width in the current study, and I guess this should be at the heart of the story.

 Line 466-469: I guess the authors need to speculate on a reason here. Would there be a reason (in the environment, in the available prey species) why such expected patter is not observed?

 Line 474: I suggest to replace 'changes' with 'preferences'.

 Line 477: I missed a clear explanation on why the community will be 'forced' towards new trophic linkages. This needs to be explained (or made more clear in case I missed it), as this is a rather strong statement.

Author Response

Thank you for your review. We have addressed each of their concerns as outlined below (Please see the attachment).

Line 27: species name should be in italic

Authors: Done

Line 120: Polychaetes can be written with small 'p': polychaetes

Authors: Done (Line 139)

Line 166: it is not clear whether 15 individuals were flipped over, or whether enough individuals were flipped over until 15 observations on stomachs were recorded.

Authors: A total of 15 individuals were flipped over to analyze stomach contents. The information was added (see lines 177-178).

Line 209-210; it is unclear what is meant with 'the level of spread'? This needs to be clarified.

Authors: The phrase was removed

In addition, I am surprised to see a combination of MDS (wich is generally a NON-metric multidimensional scaling in PRIMER). The MDS algorithm is different from the PERMANOVA algorithm. I suggest to use PCO in PRIMER, as this visualisation technique uses an algorithm that is more similar to PERMANOVA.

Authors:  We remove MDS analyses. In relation to doing PCO we thought It is not necessary because adding other analyses would show an over-analysis of the data. Differences and tend among sites are shown with PERMANOVA and different stable isotope analyses.

 Line 214: I find it strange to do spend a lot of energy (and probably money) to collect stable isotope values of organisms from 3 different regions, and then transform them. The authors could go for an alternative approach and work with the real data, and provide (or speculate) on the reasons for the outliers. Often, there is valuable ecological information in such values.

Authors: The values were transformed to PERMANOVA and MDS. For the rest of the analyses (Isotopic niche, simmr model and Layman metrics), the values were not transformed.

 Line 235: I believe that the correct notation for SEAC is with the "C" in subscript. Please check notation for SEAB as well.

Authors: SEAC is a correct notation (both notations are correct).

 3.1. This section is confusing, and I have the feeling that is over-analysed. I agree that it makes sense to check if the 13 C values, and the 15 N values are different between regions. Hence, a PERMANOVA for 13C, and a PERMANOVA for 15N with a visualisation by box-whisker plots would be sufficient. The MDS, based on 2 variables is not needed (and doesn't make sens, as 2 variables can be visualised using a x-y plot?

Authors: We remove MDS in order to avoid an over-analysis.

 Line 263. I don't understand why there would be 'more spatial structure' in the values from South Bay. The graph suggests that the values from South Bay are different from the values recorded in the other 2 locations.

Authors: We removed the paragraph and MDS.

 Line 264-266: this can be written easier.? I guess the bottom line is that the pairwise tests indicated that the values in Marian Cove are significantly lower than in the other 2 areas?

Authors: We reworded the text following the reviewer advice (see lines 300-308)

 Line 268-271: A rather lenthy description to mention that the 15N values are significantly different between regions. I actually suggest to bring all the results of the PERMANOVA (general and pairwise test) in a table, and describe the graphs. This will shorten the text, and bring clarity for the reader.

Authors: We reworded the text (see lines 300-308). The results of Permanova was added in a table (see line 308)

 Line 291 - 293: this is difficult to see in the figure. It would make more sense to have panels per 'index', and three values (for each region) per panel. Then, the reader doesn't need to check different panels for each 'index'.

Authors: Done

 Line 300: Table 2?

Authors: This was clarified in the text (see line 377)

 Line 332-333: this approach is not described in the Material and Methods? It is a linear regression, hence should be mentioned there.

Authors: Done (see lines 293-294)

 Line 336: this is actually discussion (and the authors overlooked the possibility that different prey items can have similar 13 C values)

Authors: We reworded the text (see lines 422-425). The possibility that different prey items can have similar 13C values is already suggested in the text: “Unexpectedly, this shift was not observed in individuals from South Bay, possibly reflecting uniform foraging strategy regardless of age or a homogeneous supply of basal resources” (Lines 596-605).

Line 426: Another explanation can be 'no specialisation at all', and hence the capacity to feed on very different food items?

Authors: The reviewer is right. Odontaster validus is a generalist specie. However, our study suggest that this generalist species can also enclose specialized individuals. We reworded the text to clarify this issue (see lines 538-539 and 577-578).

 Line 429-430: this needs some more explanation. Why are these new trophic linkages, and why is this very important?

Authors: We reworded the text (see lines 541-545)

 Line 445-447: it is nice to see that there are other areas with lowered isotopic niche width, and an associated explanation. However, there is no explanation provided for the lower isotopic niche width in the current study, and I guess this should be at the heart of the story.

Authors: The explanation is already in the text (see lines 566-571). We added more information in the lines 488-489.

 Line 466-469: I guess the authors need to speculate on a reason here. Would there be a reason (in the environment, in the available prey species) why such expected patter is not observed?

Authors: We added more information (see lines 511-514). The speculation of this issue is already in the text (see lines 577-578).

 Line 474: I suggest to replace 'changes' with 'preferences'.

Authors: Done (see line 613)

 Line 477: I missed a clear explanation on why the community will be 'forced' towards new trophic linkages. This needs to be explained (or made more clear in case I missed it), as this is a rather strong statement.

 Authors: We added more information (see lines 541-545)

Reviewer 2 Report (New Reviewer)

This manuscript is an ecological article about feeding ecology of Odontaster validus in the West Antarctic Peninsula, based on their δ13C and δ15N values. This manuscript constitutes a reasonably interesting piece of work that fits well in the scope of this journal as aspects of identifying the isotopic niche of O. validus among three systems with marked environmental differences associated with physical disturbances. However, although the theme is interesting, there are some problems in the manuscript.

1) Authors presented the results on size-based related shifts in δ13C and δ 15N values of O. validus. However, there was a lack of explanation for this part. Especially, this ms needs to explain why there is a regional difference related with the environmental factors. 

2)  What is Figure 7C? (Line 368). In particular, did authors do the analysis of stomach contents only in Ardley Cove?  There was no explanation for the method of stomach content analysis in the Method section.

3) More importantly, in the discussion section, authors did not explain the relationships between environmental factors and isotope data, that the differences in environmental conditions (physical disturbances) among three regions may be lead to the distinction of basal resources for O. validus. This part needs more explanation.

4) Figure 3:  Authors have to add statistical values of isotopic values among three regions.

Author Response

Thank you for your review. We have addressed each of their concerns as outlined below (Please see the attachment).

1) Authors presented the results on size-based related shifts in δ13C and δ 15N values of O. validus. However, there was a lack of explanation for this part. Especially, this ms needs to explain why there is a regional difference related with the environmental factors. 

 Authors: We added more information of the relation between δ13C and δ15N values and disk radius (see lines 605-608). Regional differences related with the environmental factors are in the lines 517-533.

2)  What is Figure 7C? (Line 368). In particular, did authors do the analysis of stomach contents only in Ardley Cove?  There was no explanation for the method of stomach content analysis in the Method section.

 Authors: Figure 7C is the simmr model output. Regarding the second question, yes, the stomach content analysis was done only in Ardley Cove as additional analyses to complement this study. Unfortunately, due to logistical issues, it was not possible to do this type of sampling in Marian Cove and South Bay.

3) More importantly, in the discussion section, authors did not explain the relationships between environmental factors and isotope data, that the differences in environmental conditions (physical disturbances) among three regions may be lead to the distinction of basal resources for O. validus. This part needs more explanation.

 Authors: Relationship between environmental factors and isotope data is already in the manuscript (see lines 517-533). However, it is difficult to add more information because basal resources of stable isotopes analyses are limited in the area of study.

4) Figure 3:  Authors have to add statistical values of isotopic values among three regions.

Authors: This information was added in the text (see lines 300-308)

Reviewer 3 Report (New Reviewer)

Generally interesting and valuable work, but the structure and presentation are partly not so clear.
First of all, it's not clear why authors add "Simple summary" to Abstract. Nothing special in this section, it can be simply included into Abstract.  From the other side there are a very few keywords, lacking very important parts of the work - feeding ecology, feeding strategy and trophic changes (due to global warming).

There are also some misunderstandings in figures. It is probably a misprint on figure 1 as upper panel indicates C and lower - A and B (opposite to written by authors). On figures 4 and 5 the scale on y-axis is definitely too small - in some cases it's difficult to catch the distance between black circles and red crosses. Also, some careless legend on figure 8 - it can't be 25th or 75th quartiles, but either 25% nad 75%, or 1 and 4 quartiles (line 390). 

Author Response

Thank you for your review. We have addressed each of their concerns as outlined below (Please see the attachment).

First of all, it's not clear why authors add "Simple summary" to Abstract. Nothing special in this section, it can be simply included into Abstract.

Authors: We have included a simple summary because this is a requirement of Biology Journal

From the other side there are a very few keywords, lacking very important parts of the work - feeding ecology, feeding strategy and trophic changes (due to global warming).

Authors: We have added more keywords including (see lines 52-53)

There are also some misunderstandings in figures. It is probably a misprint on figure 1 as upper panel indicates C and lower - A and B (opposite to written by authors).

Authors: This was clarified in the manuscript (see lines 120-121)

On figures 4 and 5 the scale on y-axis is definitely too small - in some cases it's difficult to catch the distance between black circles and red crosses.

Authors: It is not possible to change the y-axis, because range values of individuals from Ardley Cove are higher than values from other sites.

Also, some careless legend on figure 8 - it can't be 25th or 75th quartiles, but either 25% nad 75%, or 1 and 4 quartiles (line 390).

Authors: Done (see line 499)

This manuscript is a resubmission of an earlier submission. The following is a list of the peer review reports and author responses from that submission.

Round 1

Reviewer 1 Report

The manuscript, “Contrasting responses in the feeding ecology of Odontaster validus under different environmental conditions in West Antarctic Peninsula”, presents a study examining the diet of the cushion star investigating differences in feeding ecology associated with context-specific of habitat. This paper is a valuable contribution to Antarctic science, however, improvements in terms of grammar, and the conclusion, as well as figures should be made before publication. The conclusion needs strong topic sentences for each paragraph, particularly for the first two subsections.

The paragraph after the aims of the study does not fit. Please move above or eliminate.

The red star in inset A, B, C of Figure 1 is too pixelated and should be fixed. It is also not mentioned in the figure legend.

Figure 4: the red star is too pixelated

Figure 5: red crosses are difficult to see

Keyword suggestion: sea star

Minor edits:

Line 24: should say ‘species’

Line 31: should say ‘species’

Line 32: extra space in between sentences

Line 52: should say, “allowing individuals to forage over a wide range of prey”

Line 55: should say ‘species’

Line 63: or actually consume

Line 64: as ‘this method’ provides, is needed here

Line 76: Odontaster should be spelled out to begin a sentence and this sentence should read, Odontaster valius is highly dependent (scientific names are always singular)

Line 82: should say ‘species’

Line 83: needs a period after reference 16.

Line 88: quantitatively “describe”, not described.

Line 91: aims, change to past tense to keep consistent

Line 93: quantitatively “describe”, not described.

Line 100: should say ‘species’

Line 125: extra space between sentences

Line 154: correct to suction dredge sampling, remove “a”

Line 160: how did the authors know they were adults? Did they use a standard measurement?

Line 177: Is the word tegument appropriate here? I’m not sure what this word refers to. Same at line 186.

Line 188: give manufacturer and country.

Line 312: omit “had”

Line 335: should be “habitats shape”

Line 340: should say ‘species’

Reviewer 2 Report

Comments can be found in the attached document.

I think the paper needs a lot of work and to be put in a totally different context.
The authors suggest 'responses' to environmental changes linked to global changes with no use of a reference.

This study is descriptive and should be published as a descriptive paper and/or a baseline of trophic information for O. validus in this region.

I understand that there is a lot of analytical work and challenging sampling design but it is my conviction that the results should be presented in a different context.

Reviewer 3 Report

The submission by Zenteno-Devaud et al. compared the feeding ecology of red sea star Odontaster validus among three different habitats in West Antarctic Peninsula. This study reported that the red sea star had flexible foraging strategy under markedly different environmental conditions and isotopic niche expansion. The findings are interesting albeit major revision is needed before a re-submission, especially the Introduction and Discussion. Additionally, the following specific points should be addressed by the authors:

Abstract:

Line 28. between? or among? Correct this.

Introduction:

Line 50-55. I suggest remove this sentence to the 3rd paragraph, in which you introduced the stomach contents analysis.

Line 55. You have stated that O. validus might be a “keystone species,” why? Can you explain it and add some references from the literature?

Line 82-86. ‘Of dietary changes responses to marine ice loss, we know little [16] What we do know has focused on isotopic niche width in sites with different availability and diversity of resources and has neglected other pathways of quantification such as relative contribution of potential preys and trophic structure.’ The meaning is confusing, and need further clarify. Please rewrite this sentence.

Line 95-104. These paragraphs stated the feeding behaviors and ecology of the red sea star, In my opinion these parts are not well structured, and should not behind your hypothesis. I suggested you reorganize your structure of the Introduction, better to arrange the hypothesis part at last.

Materials and Methods

Line 141, 151 and 165. The subtitles should use unified format.

Line 197. Check the subscript in your equation.

Lin 198-199. 13C, 15N, N2? Correct these mistakes.

Line 243-246. The model used in this Manuscript is Simmr, needless to mention the SIAR. Like “…, we used the upgrade function Simmr from the package Stable Isotope Mixing Models within R, which contains a slightly more sophisticated mixing model that allows realistic assessments and discerning source contributions to a mixture.” I suggest use Simmr rather than SIAR in the whole MS, also in the figure captions.

Results

Line 256. ‘P’ should be in italic. Check this issue throughout the MS.

Line 274. Figure 4A is the results of SEAC results? Clarify it and re-word the caption of Figure 4. Additionally, the picture of red sea star (figure 4B) is vague, better to provide a high quality one.

Line 321. You have stated that Simmr was used in your trophic analysis in M&M, you should use Simmr output rather than SIAR. Simmr is a more reliable model and has been widely applied in trophic analysis than SIAR. Four potential food sources of the O. validus has been chosen in the MS, could you explain why these species are representative?

Line 327. The Isospace map of O. validus and their food sources should be added into the MS.

Line 328-333. Please provide the number of samples used and the corresponding isotope ratios of your potential food sources of the O. validus in a table, which are important perquisites before conducting Simmr trophic analysis. Clarify this.

Line 329. Simmr model rather than SIAR model.

Discussion

Line 334. The Discussion part is not well structured in my opinion. Subtitles may be not applicable. I suggest reorganize this part on the basis of Results and revise your language.

Line 344, 365 and 377. Subtitles may be not applicable.

Line 335 in the agreement with? or in agreement with? Revise your language.

Line 379-383. ‘This difference was also influenced by Layman metric (wider NR and CR), suggesting that these individuals forage over a range of different habitats and prey of different trophic levels, resulting in more individual isotopic differences and reflecting their generalist and opportunist status in a region characterized by a broad range of available organic matter to upper trophic consumers.’ Rephrase, not clear the meaning.

Line 417-421. ‘A similar 417 pattern was observed in the inner Ezcurra Inlet Admiralty Bay, King George Island, where 418 low resource availability linked to higher turbidity induce trophic niche constriction and 419 interspecific resource segregation, which in turn could be reflecting a mechanism for com-420 petition avoidance in a resource-limited system.’ Simply this sentence, and clear the meaning.

References:

Line 487, 517, 518, 588, 600, 602, et al: Check and use the standard format in all references.